# Immune Checkpoint Inhibitors Combined with Targeted Therapy: The Recent Advances and Future Potentials

**DOI:** 10.3390/cancers15102858

**Published:** 2023-05-22

**Authors:** Bin Li, Juan Jin, Duancheng Guo, Zhonghua Tao, Xichun Hu

**Affiliations:** 1Department of Breast and Urologic Medical Oncology, Fudan University Shanghai Cancer Center, Shanghai 200032, China; 16301050247@fudan.edu.cn (B.L.); medjinjuan@shca.org.cn (J.J.); guoduancheng@shca.org.cn (D.G.); 2Department of Oncology, Shanghai Medical College, Fudan University, Shanghai 200032, China

**Keywords:** immune checkpoint inhibitors, targeted therapy, combined therapy, solid tumors, PD-1/PD-L1

## Abstract

**Simple Summary:**

Immune checkpoint inhibitors (ICIs) are the most mature treatment of immunotherapy and have changed the mode of cancer treatment. Recent studies showed that combining ICIs with targeted drugs is a potential strategy in some tumors, while in other tumors, this combination increases toxicity but does not improve efficacy. This review first comprehensively covers the current status of studies in the combination of ICIs and targeted drugs in the treatment of solid tumors, involving the underlying mechanisms, clinical effects, side effects, and potential predictive biomarkers, and provides a perspective for future directions and potential therapeutic strategies of immunotherapy in solid tumors.

**Abstract:**

Immune checkpoint inhibitors (ICIs) have revolutionized the therapeutic landscape of cancer and have been widely approved for use in the treatment of diverse solid tumors. Targeted therapy has been an essential part of cancer treatment for decades, and in most cases, a special drug target is required. Numerous studies have confirmed the synergistic effect of combining ICIs with targeted therapy. For example, triple therapy of PD-L1 inhibitor atezolizumab plus BRAF inhibitor vemurafenib and MEK inhibitor cobimetinib has been approved as the first-line treatment in advanced melanoma patients with *BRAF*^V600^ mutations. However, not all combinations of ICIs and targeted therapy work. Combining ICIs with EGFR inhibitors in non-small-cell lung cancer (NSCLC) with *EGFR* mutations only triggered toxicities and did not improve efficacy. Therefore, the efficacies of combinations of ICIs and different targeted agents are distinct. This review firstly and comprehensively covered the current status of studies on the combination of ICIs mainly referring to PD-1 and PD-L1 inhibitors and targeted drugs, including angiogenesis inhibitors, EGFR/HER2 inhibitors, PARP inhibitors and MAPK/ERK signaling pathway inhibitors, in the treatment of solid tumors. We discussed the underlying mechanisms, clinical efficacies, side effects, and potential predictive biomarkers to give an integrated view of the combination strategy and provide perspectives for future directions in solid tumors.

## 1. Introduction

Immunotherapy has revolutionized the field of oncology. Immune checkpoint blockade therapy is arguably the most widely used immunotherapy approach in clinical practice. Immune checkpoints are molecules that maintain a balance of co-stimulatory and co-inhibitory signals on the surface of T cells and tumor cells, and tumor cells utilize immune checkpoints to escape from the attack of the immune system and induce tumor immune escape [1,2]. At present, immune checkpoint inhibitors (ICIs) have been approved for the treatment of various cancers, as monotherapy or combined with radiotherapy or chemotherapy [3].

ICI monotherapy is effective in immunologically hot tumors, such as melanoma and microsatellite instability-high (MSI-H) tumors, while the effective rate of monotherapy is low in unselected solid tumors [4,5]. The currently widely used ICIs are PD-1/PD-L1 inhibitors, which show significant clinical benefits in tumors with a positive expression of PD-L1, but the proportion of this population in some tumors is low, for example, only 20% of breast cancer patients have a positive PD-L1 expression [6]. Therefore, how to further explore the therapeutic potential by optimizing predictive biomarkers and combining them with other agents is the current focus of the study. The approved combined treatments based on ICIs are mainly with chemotherapy, but substantial side effects and limitedly improved benefits are observed in some tumors [7,8]. Alongside chemotherapy, combining ICIs with targeted agents is a crucial strategy and some clinical trials have shown significant antitumor effects of this combination.

Targeted therapy has been an indispensable part of cancer treatment with high specificity and different side effect profiles compared to chemotherapy. According to the drug type, targeted drugs are mainly classified into small molecules and monoclonal antibodies, which act on growth factors, angiogenesis receptors, signal pathways, etc. [9,10]. Preclinical studies indicated that targeted therapy might dampen immunosuppression induced by tumors, providing the potential to enhance the efficacy of immunotherapy and then produce a synergistic effect to activate the immune system [11,12,13]. The United States Food and Drug Administration (FDA) approved the combination of PD-L1 inhibitor atezolizumab with BRAF inhibitor vemurafenib plus MEK inhibitor cobimetinib for the first-line treatment of advanced melanoma patients with *BRAF*^V600^ mutations in 2020. However, in patients with *EGFR*-mutant non-small cell lung cancer (NSCLC), the clinical benefits were not achieved when PD-L1 inhibitor durvalumab was combined with epidermal growth factor receptor–tyrosine kinase inhibitor (EGFR-TKI) osimertinib, and the combination only incurred more adverse events. Therefore, we are interested in which targeted drugs combined with ICIs could produce a synergistic effect and which patients are available for the combined therapy.

In this study, we systematically reviewed the existing studies of ICIs combined with targeted therapy in solid malignant tumors to determine how to conduct the combinational treatment. We investigated the underlying mechanisms, predictive factors, and related side effects.

## 2. ICIs Used in Clinical Practice

### 2.1. PD-1/PD-L1 Inhibitors

PD-1 is mainly expressed on the membrane of activated T cells and can bind to two ligands, PD-L1 and PD-L2 [14]. PD-L1 is mainly expressed on the surface of tumor cells and immune cells, such as dendritic cells, macrophages, and activated T cells [15,16]. PD-1 binds to its ligands and then recruits cell phosphatases SHP1/2 to inhibit early T cell receptor signals. It also can inhibit the RAS-RAF-MEK-ERK and PI3K-AKT pathways to suppress the proliferation of T cells [17,18]. Therefore, inhibiting the PD-1/PD-L1 signaling can impair the inhibition of T cell activation and enhance immune responses to prevent the immune escape of tumors.

PD-1 and PD-L1 inhibitors have been widely used in cancer immunotherapy, with great developments over the years. Nivolumab and pembrolizumab are commonly used as PD-1 inhibitors, and atezolizumab, durvalumab and avelumab belong to PD-L1 inhibitors. At present, the FDA has approved PD-1/PD-L1 inhibitors for treating a wide spectrum of solid tumors, but in clinical practice, the application of PD-1/PD-L1 inhibitor monotherapy has an unsatisfactory response rate in unselected patients [19,20]. Combining chemotherapy with PD-1/PD-L1 inhibitors is a mainly used strategy nowadays, but patients who benefit from this combination treatment are still limited [21]. Other therapy strategies, such as combining PD-1/PD-L1 inhibitors with targeted therapy, radiotherapy, or other immunotherapy, have been investigated to improve the effect of PD-1/PD-L1 inhibitors and enormous clinical trials are ongoing, some of which have exhibited exciting results [14,22,23,24,25].

### 2.2. Others

CTLA-4 was the first-discovered immune checkpoint as a negative regulator of immune responses, which was mainly expressed in regulatory T cells (Tregs) [26,27]. CTLA-4 interacts with the ligands CD80 and CD86 to inhibit T-cell-related responses [28,29,30]. Accordingly, blocking CTLA-4 can enhance T cell responses in tumors [31,32,33,34,35]. The monotherapy of CTLA-4 inhibitor ipilimumab was approved by the FDA in unresectable advanced melanoma in 2011, which was the first ICI applied in clinical practice [36]. The clinical trials of ipilimumab in other solid tumors are still ongoing, but most of them have a low response rate (NCT01585987, NCT01471197).

In addition, other ICIs including LAG-3 (lymphocyte activation gene-3) inhibitors, anti-TIM-3 (T cell immunoglobulin and mucin domain-3) monoclonal antibodies, Tight (T cell immunoglobulin and ITIM domain) inhibitors, and anti-Siglec-15 monoclonal antibodies have been studied in clinical trials. Based on the preclinical data that LAG-3 and TIM-3 could work in coordination with PD-1, the combination of LAG-3/TIM-3 inhibitors with PD-1/PD-L1 inhibitors in solid tumors might be a potential strategy [37,38,39,40]. According to the latest progress, a phase II-III study, RELATIVITY-047, demonstrated that the combination of nivolumab and relatlimab (a LAG-3 inhibitor) improved progression-free survival (PFS) in advanced melanoma patients with comparable adverse events, which promoted the approval of the combination of nivolumab and relatlimab for the first-line treatment of advanced melanoma by FDA [41].

## 3. ICIs Combined with Targeted Drugs

### 3.1. ICIs Combined with Angiogenesis Inhibitors

Angiogenesis is indispensable for the growth and metastasis of solid tumors. Proliferating tumor cells need enough nutrients and oxygen, so these cells promote the formation of blood vessels nearby [42]. In the tumor microenvironment (TME), active angiogenesis may lead to immunosuppression via decreasing the abundance and function of effector immune cells. The high interstitial fluid pressure in active angiogenesis makes a barrier for effector T lymphocytes infiltration, and active angiogenesis of tumors also increases the VEGF (vascular endothelial growth factor) in the circulatory system, which suppresses the function of dendritic cells [43]. In addition, active angiogenesis of tumor upregulates the suppressive immune molecules, such as PD-L1, IL-6, IL-10, and Fas ligand, which recruit Tregs into TME and clean up cytotoxic T lymphocytes to make it difficult to kill the tumor cells [44,45]. Based on the above mechanisms, combining ICIs and angiogenesis inhibitors may get more benefits in cancer treatment. Early in 2013, it was found that combining anti-PD-1 monoclonal antibody and anti-VEGFR2 monoclonal antibody had a synergistic anti-tumor effect in a murine model with colon adenocarcinoma, suggesting that it might be an effective strategy in clinical practice [46].

Angiogenesis inhibitors include monoclonal antibodies and small molecular inhibitors, and now monoclonal antibodies have been more widely used with ICIs. Most recently, a phase III study, IMbrave 150, enrolled 336 patients with unresectable hepatocellular carcinoma (HCC) in atezolizumab plus bevacizumab group and 165 ones in sorafenib monotherapy group. The results showed that the median overall survival (mOS) and median PFS (mPFS) was 19.2 and 6.8 months in atezolizumab plus bevacizumab group and 13.4 and 4.3 months in sorafenib group (OS: HR, 0.66; 95% CI 0.52–0.85; descriptive *p* < 0.001; PFS: HR, 0.65; 95% CI 0.53–0.81; descriptive *p* < 0.001), respectively [47,48]. Based on IMbrave 150, IMbrave 050 investigating the combination of atezolizumab plus bevacizumab in HCC patients at a high risk of recurrence after surgical resection or ablation is still in the research phase [49]. Another phase III trial, ORIENT-32, had compared the combination of sintilimab (PD-1 inhibitor) and IBI305 (anti-VEGF monoclonal antibody) with sorafenib monotherapy in 571 Chinese patients with HBV-associated HCC. This study indicated that patients in the sintilimab–bevacizumab group had a longer mPFS (4.6 months) than patients in the sorafenib group (2.8 months), and the mOS was not reached in the sintilimab–bevacizumab group versus 10.4 months in sorafenib group [50]. The combination of PD-L1 inhibitor atezolizumab and bevacizumab also showed safety and observed longer PFS in PD-L1 positive subgroup of patients with advanced renal cell carcinoma (RCC) [51]; the mPFS was 11.2 months for atezolizumab plus bevacizumab versus 7.7 months for sunitinib in PD-L1-positive patients [52]. In conclusion, combining PD-1/PD-L1 inhibitors and bevacizumab as the first-line treatment can improve the curative effect in patients with advanced HCC, and FDA-approved atezolizumab plus bevacizumab for patients with advanced HCC who had not received prior systemic therapy and whether PD-L1 is positive or not in these patients. In these trials, the most common grade 3 or 4 adverse events in the combined therapy group are hypertension, increased aspartate aminotransferase, and proteinuria, which occurred more frequently in the combined treatment group than the control group.

The phase II trial of non-squamous NSCLC, Be Study, suggested that the combination of atezolizumab and bevacizumab was a potential strategy for patients with PD-L1 positive and without *EGFR/ALK/ROS1* alterations [53]. In this study, 25 out of 39 (64.1%) patients with PD-L1 positive and without *EGFR/ALK/ROS1* alterations had a partial response, and 1-year OS and PFS rates were 70.6% and 54.9%, respectively. Although the efficacy of bevacizumab plus chemotherapy is manifested in metastatic NSCLC, the addition of ICIs can further improve clinical outcomes, which has been confirmed by a phase III study (IMpower150) [54,55]. Based on the results from IMpower150, atezolizumab–bevacizumab–carboplatin–paclitaxel (ABCP) was approved by the FDA as a first-line treatment in patients with advanced non-squamous NSCLC [56]. The grade 3–4 treatment-related events occurred in 57% of patients in the ABCP group, suggesting that we should pay more attention to the side effects of this combination. In patients with metastatic colorectal carcinoma (mCRC), a phase II trial, AtezoTRIBE, showed that the addition of atezolizumab to bevacizumab plus FOLFOXIRI (fluorouracil, leucovorin, oxaliplatin, and irinotecan) was safe and improved the mPFS compared to bevacizumab plus chemotherapy (13.1 months versus 11.5 months; HR, 0.69; 80% CI 0.56–0.85; *p* = 0.012) and OS is not yet mature [57]. However, another study, BACCI, showed that atezolizumab plus bevacizumab–capecitabine only improved the PFS versus bevacizumab–capecitabine (4.4 vs. 3.3 months). Still, the OS and response rate were not significantly improved in patients with refractory mCRC [58]. Therefore, the chemotherapy agents might influence the efficacy of atezolizumab and bevacizumab. It is disappointing that a phase II study in 11 patients with advanced cervical cancer indicated that the combination of bevacizumab and atezolizumab showed a low objective response rate (ORR, 0%) and short mPFS (2.9 months) and mOS (8.9 months) [59]. As bevacizumab is approved for patients with metastatic cervical cancer by the FDA in combination with chemotherapy, chemotherapy is probably necessary for synergistic effects in this context.

In addition, the small molecular anti-vascular inhibitors are being explored in many studies. The phase III JAVELIN Renal 101 trial, in which 442 patients with advanced RCC were included in the avelumab–axitinib group and 444 in sunitinib monotherapy group, showed that the mPFS was 13.8 months with avelumab–axitinib versus 7.0 months with sunitinib in PD-L1 positive patients, and was 13.3 months versus 8.0 months in overall population, but the OS was immature [60]. In another phase III study (KEYNOTE-426), 432 advanced RCC patients were in the pembrolizumab–axitinib group and 429 patients were in sunitinib group, in which more than 50% patients in both groups had ≥1 of PD-L1 combined positive score [61]. The mOS and the mPFS was longer in pembrolizumab-axitinib group than sunitinib group (mOS: not reached vs. 35.7 months; mPFS: 15.4 months vs. 11.1 months). A phase II trial (EPOC1706) indicated that the ORR of the combinational treatment of lenvatinib (multi-kinase VEGFR inhibitor) and pembrolizumab reached to 69% in 29 advanced gastric cancer patients after heavy treatment. The mPFS was 7.1 months and the mOS was not reached [62].

In summary, the combination of ICIs with angiogenesis inhibitors can produce synergistic effects among the advanced tumors in which angiogenesis inhibitors are approved, including HCC, RCC, NSCLC, and CRC, which has been listed in Table 1. Additionally, the ongoing trials are summarized in Appendix A. For advanced CRC and NSCLC, chemotherapy and angiogenesis inhibitors are the standard treatments. Therefore, chemotherapy is still indispensable in the combinational treatment of ICIs and angiogenesis inhibitors in these cancers. Combining FOLFOXIRI plus bevacizumab is the first-line treatment for advanced CRC, and bevacizumab combined with carboplatin plus paclitaxel is also a first-line treatment in NSCLC. Therefore, chemotherapy is indispensable in the combinational treatment of ICIs and angiogenesis inhibitors in these cancers, and there is no dose adjustment of the chemotherapy agents. The new adverse events were not observed in combined therapy, but we should pay more attention to neutropenia and anaphylaxis caused by platinum-based chemotherapy drugs and give active pre-treatment, such as using granulocyte colony-stimulating factor and glucocorticoids.

### 3.2. ICIs Combined with EGFR Inhibitors or HER2 Inhibitors

The EGFR (epidermal growth factor receptor) family belongs to protein tyrosine kinase groups, which play a general role in oncogenesis. In normal tissues, EGFR regulates the development of epithelial tissue and signal transduction, while EGFR may mutate or be overexpressed in tumor cells leading to the proliferation of tumor cells and metastasis by activating the EGFR-PI3K/AKT pathway and stimulating some matrix metalloproteinases such as MMP1 and MMP9 [63]. At present, EGFR-TKIs are the first-line treatment for advanced NSCLC patients with *EGFR* mutations but acquired resistance eventually emerges [64]. HER2 (human epidermal growth factor receptor 2) is a non-ligand-binding member of the EGFR family, which activates through heterodimerization with other EGFR family members [65]. It is certificated that the overexpression of HER2 was 15–30% in breast cancer and 10–30% in gastric/gastroesophageal cancer, associated with a poor prognosis [66,67]. HER2 inhibitors are cornerstones of therapy in patients with HER2-overexpressed breast cancer and gastric cancer. A preclinical study has shown that in murine NSCLC models, the activation of the EGFR pathway is associated with immunosuppression [68]. The activation of the EGFR pathway could increase the expression of PD-L1 via p-ERK1/2/p-c-Jun, upregulate the abundance of Tregs and M2 macrophages in TME and induce the excretion of some immunosuppressive molecules such as IL-6 and TGF-β1, which suggest that patients may benefit from combining immunotherapy with EGFR inhibitors [69].

Although PD-1/PD-L1 inhibitors have shown promising results in lung cancer, they may be limited in patients with *EGFR* mutations. The potential of combining ICIs with EGFR inhibitors in NSCLC with *EGFR* mutations is disappointing to date due to the lower response rate and higher number of toxicities [70]. In 2019, the results of the phase I/II trial, KEYNOTE-021, showed that PD-1 inhibitor pembrolizumab in combination with gefitinib for advanced NSCLC with *EGFR* mutations was not feasible because of the high occurrence of liver toxicity (71.4%) [71]. Another cohort of this trial also evaluated the efficacy of combining pembrolizumab and erlotinib. The mPFS was 19.5 months and the ORR (41.7%) was not improved compared with previous erlotinib monotherapy. The most common grade 3 or 4 adverse events were skin reactions, diarrhea, and hypothyroidism and there were no grade 5 adverse events. Another phase II trial, LUX-Lung IO, was not hopeful and ended early, which investigated the combination of pembrolizumab and afatinib in advanced squamous cell lung carcinoma. A phase III trial, CAURAL, evaluated 29 patients with *EGFR*^T790M^-positive NSCLC to treat osimertinib (an EGFR inhibitor) plus durvalumab versus osimertinib monotherapy [72]. The ORR was 64% in the osimertinib–durvalumab group and 80% in osimertinib group, while the median duration of response (DOR) was 21.4 months in the osimertinib–durvalumab group versus 17.5 months in the osimertinib group. It was terminated early because of the increased interstitial lung disease incidence. An observational study including 20,516 patients with NSCLC investigated the incidence of interstitial pneumonitis in patients treated with EGFR-TKI, nivolumab, and EGFR-TKI plus nivolumab [73]. There were 70 patients treated with EGFR-TKI plus nivolumab, of which 18 patients (25.7%) developed interstitial pneumonitis and 15 of these 18 patients were treated with EGFR-TKI after nivolumab. It was found that nivolumab increased related interstitial pneumonitis by increasing the odds ratio from 1.22 to 5.09.

Combining ICIs and HER2-targeted inhibitors also have been investigated. A phase I trial, CCTG IND.229, showed that combining PD-L1 inhibitor durvalumab with trastuzumab in HER2-positive metastatic breast cancer showed no responses in patients, with an mPFS of 1.35 months [74]. Another study, PANACEA, also demonstrated a pessimistic outcome of the combination of pembrolizumab and trastuzumab in patients with trastuzumab-resistant HER2-positive breast cancer with an ORR of 15% in PD-L1 positive patients [75]. However, a phase II trial (KATE2), in which 202 patients with HER2-positive advanced breast cancer were randomized into an atezolizumab plus T-DM1 (trastuzumab emtansine) group and a T-DM1 monotherapy group, showed an mPFS of 8.2 months in a combined therapy group versus 6.8 months in T-DM1 alone group in the intention-to-treat population. In a PD-L1-positive subgroup, the mPFS was 8.5 months in the combined therapy group and 4.1 months in the T-DM1-alone group [76]. A phase Ib/II trial (CP-MGAH22–05) demonstrated an ORR of only 18.48% in 95 HER2-positive gastro-esophageal adenocarcinoma patients treated by Margetuximab (HER2 inhibitor) plus pembrolizumab [77]. Notably, among 88 patients with available microsatellite stable status, only one patient (1%) was MSI-H [77]. The phase III trial, KEYNOTE-811, showed that combined pembrolizumab and trastuzumab plus chemotherapy significantly improved the ORR in patients with metastatic HER2-positive gastric or gastro-esophageal junction adenocarcinoma [78]. The ORR was 74.4% in the pembrolizumab group versus 51.9% in the placebo group. In the pembrolizumab group, 15 patients (11.3%) had a complete response, while in a placebo group, only 4 patients (3.1%) had a complete response.

The existing results suggested that no evident synergistic effects of the combination of ICIs and anti-EGFR inhibitors in *EGFR*-mutated lung cancer were observed (Table 1), and this combined strategy only induced a lower response rate and more toxicities. The combination of ICIs with traditional HER2 inhibitors such as trastuzumab has poor efficacy, while ICIs combined with HER2-targeted antibody-drug conjugates (ADCs), such as trastuzumab emtansine or the novel ADC T-DXd, showed an effective response in HER2-positive PD-L1 positive breast cancer or metastatic HER2-positive gastric or gastro-esophageal junction adenocarcinoma, which still needs to be further confirmed. Further, for tumors with HER2 overexpression, the addition of chemotherapy might be crucial for the efficacy of the combination of ICIs and anti-HER2 therapy. Accordingly, in tumors with HER2 overexpression, the combinational efficacy needs to be further explored (Appendix A).

### 3.3. ICIs Combined with Poly (ADP-Ribose) Polymerase Inhibitors

Poly ADP-ribose polymerase (PARP) is a key molecule of single-stranded DNA break repair. PARP inhibitors can block the repair of base resection, leading to the failure of single-stranded DNA breaks repair and eventually resulting in DNA double-strand breaks. DNA double-strand breaks mainly rely on homologous recombination repair (HRR) to repair and applying PARP inhibitors in patients with defective HRR function may lead to synergistic lethality [79]. BRCA1/2 are important molecules in the HRR pathway and *BRCA1/2* mutations lead to defective HRR function [80]. PARP inhibitors were approved by the FDA in advanced ovarian cancer and metastatic breast cancer with germline *BRCA1/2* mutations. Recent studies have shown that PARP inhibitors were involved in anti-tumor immunity, suggesting that combining PARP inhibitors with ICIs will be a potential strategy [13,81]. It was indicated that PARP inhibitors induced the accumulation of DNA fragments which could activate the DNA sensing cGAS-STING pathway and promote the production of type I interferon to activate CD8+ T cells. PARP inhibitors also increased the abundance of dendritic cells to present antigens in TME. Further, the use of PARP inhibitors may induce more DNA damage and more neoantigens and upregulate the expression of PD-L1 via interferon, which may enhance the response to ICIs [82].

A phase I/II open-label study (MEDIOLA) suggested that the application of olaparib (PARP inhibitor) and PD-L1 inhibitor durvalumab in 30 patients with *BRCA1/2*-mutated metastatic breast cancer was tolerable and mOS were 21.5 months at a median follow-up of 19.8 months [83]. Another phase I/II trial, TOPACIO, showed an ORR of 21% in 55 metastatic triple-negative breast cancer (TNBC) patients treated by the combination of niraparib (PARP inhibitor) and pembrolizumab and the ORR was higher in *BRCA1/2*-mutant patients than in *BRCA1/2*-wt (wt: wide type) patients (47% vs. 11%) [84]. Notably, the cohort of recurrent platinum-resistant ovarian cancer in TOPACIO indicated that the ORR among patients with or without *BRCA1/2* mutations or homologous recombination deficiency (HRD) seemed to be similar (*BRAC1/2*-mutation: 18%, *BRCA1/2*-wt: 19%; HRD positive: 14%, HRD negative: 19%) [85]. In addition, a part of the phase I/II trial (NCT02484404) showed that durvalumab combined with olaparib had potential efficacy in prostate cancer with deficient DNA damage repair function. This study already enrolled 17 patients, and the 12-month PFS rate was 83.3% in patients with DNA damage repair gene mutations and 36.4% in patients without mutations [86]. Recently, a phase II trial investigated the clinical effect of combining durvalumab and olaparib plus paclitaxel (DOP) neoadjuvant chemotherapy in patients with high-risk stage II/III HER2-negative breast cancer [87]. Among 372 patients, 73 patients received DOP therapy and 37% of them had a pathologic complete response, while 299 patients did standard paclitaxel and 20% of them had a pathologic complete response. Immune-related adverse events were more common in the DOP group, including hypothyroidism, pneumonitis, adrenal insufficiency, thyroiditis, and pancreatitis. JASPER, a phase II trial investigating pembrolizumab with niraparib as the first-line treatment in advanced NSCLC patients without *EGFR* mutations and *ROS1*/*ALK* translocations, showed that the ORR was 56.3% versus 20% in tumors with PD-L1 tumor proportion score (TPS) ≥ 50% and TPS < 50%, respectively, with similar side effects in those two groups [88]. To better explore the safety and efficacy of ICIs with PARP inhibitors, there are several phase II/III clinical trials ongoing (NCT03602859, NCT03642132, NCT02571725).

According to the current evidence, the combination of ICIs and PARP inhibitors has the potential efficacy in patients with metastatic breast cancer, ovarian cancer, prostate cancer, and NSCLC. It is observed that this combination is more effective in patients with metastatic TNBC or prostate cancer carrying *BRCA1/2* mutations or HRD, while the curative effect is similar between ovarian cancer patients with or without *BRCA1/2* mutations or HRD [84,85,86].

### 3.4. ICIs Combined with MAPK/ERK Signaling Pathway Inhibitors

The MAPK/ERK signaling pathway plays an important role in cell proliferation, differentiation, and apoptosis. The excessive activation of the MAPK/ERK pathway can cause the growth or metastasis of tumor cells [89]. RAF-MEK-ERK cascade is a classic pathway of MAPK/ERK, and it can be activated by RAS, and each component in this pathway is the target for cancer treatment [90]. In vivo and in vitro studies, applying a BRAF-targeted inhibitor (one of MAPK/ERK signaling inhibitors) could increase the expression of tumor antigens and the abundance of tumor-infiltrating immune cells [91]. In 2011, FDA approved the first targeted drug, vemurafenib (BRAF inhibitor), for the treatment of metastatic melanoma with *BRAF*^V600E^ mutation. Then, cobimetinib (MEK inhibitor), trametinib (MEK inhibitor), and dabrafenib (BRAF inhibitor) was approved by FDA to treat melanoma with *BRAF*^V600^ mutations, and trametinib and dabrafenib could also treat NSCLC with *BRAF*^V600E^ mutation. However, it was found that MAPK/ERK signaling pathway inhibitors just induced temporary responses, and how to extend the duration of the benefit is the focus of current studies on tumors with *BRAF*^V600^ mutations. It was shown that MAPK/ERK signaling inhibitors increased the expression of tumor antigen and human leukocyte antigen, enhanced T cells infiltration in TME, reduced immunosuppressive cytokines, and increased the expression of PD-L1 [92,93]. In addition, MAPK/ERK signaling inhibitors could protect CD8+ T cells and decrease the interaction between tumor cells and M2 macrophages [93].

A phase I/II clinical trial, KEYNOTE-022, demonstrated that in melanoma patients with poor prognostic factors, the mPFS was 16.0 months in the BRAF-MEK inhibitor (dabrafenib and trametinib) and pembrolizumab group versus 10.3 months in dabrafenib and trametinib group [94]. Furthermore, a phase III study, IMspire150, investigating atezolizumab plus vemurafenib (RAF inhibitor) and cobimetinib (MEK1 inhibitor) in metastatic melanoma with *BRAF*^V600^ mutations, showed that the mPFS was 15.1 months in the combined therapy group, while the mPFS was 10.6 months in the control group (placebo plus cobimetinib and vemurafenib) [95]. During the statistical period, 205 patients died, which was 36% (93/256) in a combined therapy group, lower than the control group (43%, 112/258). However, the median DOR in the combined therapy group was 21.0 months, much longer than the control group (12.6 months). However, COMBI-i, a phase III trial, showed that combining spartalizumab (PD-1 inhibitor) and the BRAF-MEK inhibitor (dabrafenib and trametinib) for *BRAF*^V600^-mutant advanced melanoma patients induced modest improvement of mPFS than the BRAF-MEK inhibitor group (16.2 months vs. 12 months), but there were high rates of grade 3 adverse events (55% vs. 33%) [96]. In the combined therapy group, there was a higher occurrence of increased liver enzymes, pneumonitis, rash, and hyperthyroidism, which might relate to immunotherapy. Based on the results from IMspire150, the combination of atezolizumab with vemurafenib and cobimetinib was approved by FDA for the first-line treatment of advanced melanoma patients with *BRAF*^V600^ mutations [97], and a combination of ICIs with MAPK/ERK signaling inhibitors were also investigated in other solid tumors, such as mCRC, NSCLC, and TNBC. A phase III trial investigated the clinical effect of combining atezolizumab with cobimetinib versus regorafenib in mCRC, in which 54% of patients in each group were with *RAS* mutations, but no improvement of OS in the combined group compared to the monotherapy group was observed, which may be due to the microsatellite-stable status in most patients [98].

The efficacy of the combination of ICIs and MAPK/ERK targeted drugs showed heterogeneity in treating melanoma. In melanoma patients with *BRAF*^V600^ mutations, combining atezolizumab plus vemurafenib and cobimetinib could bring a more lasting curative effect [95], while the efficacy of another triple combination therapy (spartalizumab plus dabrafenib and trametinib) is not significant [96]. Although the drug targets are the same, different drugs and the design of clinical trials may affect the clinical outcomes. It needs to be further confirmed whether it is valid in other solid tumors with mutations on MAPK/ERK signaling pathway.

### 3.5. ICIs Combined with Hormone Receptor Inhibitors

Hormone receptors play a crucial role in some specific cancers, such as breast cancer, prostate cancer, and ovarian cancer. Androgen receptor (AR), estrogen receptor (ER), and progesterone receptor (PR) are important hormone receptors, and their inhibition of them can improve the clinical outcomes in patients with hormone-dependent tumors, such as endocrine therapy in breast cancer and using apalutamide (AR inhibitor) for castration-sensitive prostate cancer [99]. A study demonstrated that the estrogen pathway promoted the proliferation of Tregs and upregulated the expression of PD-L1 on tumor cells, which modulated the signaling between immune cells and tumor cells to mediate tumor immune evasion [100]. Another recent study verified that the inhibition of AR enhanced the function of CD8+ T cells by increasing the expression of IFN-γ and then improved the sensitivity of PD-1/PD-L1 inhibitors [101]. Based on this preclinical evidence, combining ICIs and hormone receptor inhibitors has also been explored nowadays. Quite a few ongoing clinical studies are concentrating on prostate cancer (NCT03338790, NCT04926181) and hormone receptor-positive/HER2-negative (HR+/HER2−) advanced breast cancer (NCT03393845). However, the tumor-infiltrating lymphocytes, TMB (tumor mutation burden), and PD-L1 in HR+/HER2− breast cancer are all at a low level, which suggests that ICIs may get a poor response in these patients [102,103]. All in all, further exploration is needed on the role of hormone receptors in TME, and the efficacy and safety in the combination of ICIs with hormone receptor inhibitors are still the focus we need to pay attention to.

### 3.6. ICIs Combined with Other Targeted Drugs

CDKs (cyclin-dependent kinases) are important factors in the regulation of the cell cycle, which provide targets for treatment in cancer [104]. Highly selective CDK4/6 inhibitors had significant efficacy and tolerant toxicity in hormone receptor positive breast cancer and were in an exploratory status in other tumors [105,106]. The inhibition of CDK4/6 could trigger anti-tumor immunity, and a preclinical study showed that combining CDK4/6 inhibitor plus PI3K inhibitor with ICIs in TNBC murine model induced a long-lasting tumor regression [107,108]. The clinical trials on advanced breast cancer and NSCLC are ongoing (NCT03573648, NCT02779751).

ALK (anaplastic lymphoma kinase) is a receptor tyrosine kinase, which plays an important role in lung cancer. ALK inhibitors just like alectinib and lorlatinib are approved by FDA for ALK-positive NSCLC patients, but resistance finally followed [109]. To expand the indication and efficacy, there are studies exploring ALK inhibitors in combination with ICIs. A phase I study of 13 patients treated with nivolumab and crizotinib (an ALK inhibitor), showed that 5 patients (38%) had a partial response and 5 patients (38%) had severe hepatic toxicity, with the death of 2 patients [110]. This trial was closed early because of the hepatic toxicity, which implied that the adverse events of the combination of ICIs and ALK inhibitors were serious.

The PI3K/AKT/mTOR signaling pathway is one of the most common abnormal activation signaling pathways in tumors [111]. PI3K/AKT/mTOR inhibitors can not only act on tumor cells but also work on immune cells and TME, which can improve the capacity of tumor immune surveillance, so combining ICIs and PI3K/AKT/mTOR inhibitors may promote the efficacy and reduce the resistance [112]. At present, related clinical trials are ongoing about urothelial cancer (NCT03980041), metastatic melanoma (NCT03131908), head and neck squamous cell carcinoma (NCT03735628) and colon cancer (NCT03711058).

## 4. Discussion

In recent years, both ICIs and targeted therapy are important aspects of cancer treatment. Our study comprehensively reviewed the preclinical and clinical studies for the efficacy and toxicity of the combination of ICIs and targeted therapy in cancer to clarify the directions deserving further investigation and hurdles to avoid.

### 4.1. The Current Effective Strategies of Combined Therapy

Clinical trials have demonstrated that combining PD-1/PD-L1 inhibitors with angiogenesis inhibitors as the first-line treatment for advanced HCC and RCC significantly improves the OS and PFS and using PD-L1 inhibitor atezolizumab with RAF-MEK inhibitors (vemurafenib plus cobimetinib) in patients with *BRAF*^V600^-mutant metastatic melanoma has a longer clinical remission [47,50,60,61,95]. Combining PD-1/PD-L1 inhibitors with PARP inhibitors in breast cancer with germline *BRCA1/2* mutations and ovarian cancer has acceptable toxicity and potential efficacy [84,85]. Notably, combining ICIs and angiogenesis inhibitors based on chemotherapy has a better effect than adding ICIs or angiogenesis inhibitors alone in NSCLC and refractory mCRC, and it also shows the efficacy of combining ICIs with PARP inhibitors plus chemotherapy in HER2-negative breast cancer [54,58,87]. Chemotherapy might be indispensable for the synergistic effect of ICIs and targeted therapy in some cancers. The combination of ICIs and HER2-targeted ADC may be an innovative strategy for cancer treatment on account of the great benefits of atezolizumab plus T-DM1 in PD-L1-positive, HER2-positive advanced breast cancer and the rapid development of diverse anti-HER2 ADCs [76]. The effective combination strategies have been summarized in Figure 1. There is a new triple treatment of anti-PD-1 antibody camrelizumab combined with angiogenesis inhibitor apatinib plus PARP inhibitor fuzuloparib, which showed safety and preliminary antitumor activity in patients with TNBC [113]. It suggests that combining ICIs and agents targeting more than one signaling pathway is also a feasible combination therapy in the future.

In addition, other immunotherapies may play an important role in the future. Chimeric antigen receptor T cell (CAR-T) immunotherapy, which kills tumor cells by extracting T cells from the patient’s body and re-inputting them after genetic engineering methods, had excellent clinical efficacy in hematological malignancies and solid tumors. It might be combined with ICIs or costimulatory molecules to reverse the immunosuppressive microenvironment [114,115]. Natural killer (NK) cell therapy, which used natural immune cells of patients, mainly including NK cell infusion and CAR-NK cell, is combined with cytokines, ICIs, and CAR-T therapy in clinical practice [116,117]. A study showed that the combination of CDK4/6 inhibitors and MAPK inhibitors might promote NK cell surveillance via the secretion of cytokines, such as TNF-α, which suggested the potential of combined therapy [118]. Tregs induce immune escape of tumors and should be inhibited to enhance the anti-tumor effect by inhibiting the molecules highly expressed in Tregs [119,120]. It was shown that the VEGF-VEGFR2 axis involved in the accumulation of immature Tregs, tyrosine kinase inhibitors might inhibit TCR signaling for Tregs survival and function, and the use of PI3K inhibitors in the mouse model may induce immunosuppression via Tregs, which suggests the possibility of combining Tregs anti-tumor therapy with targeted therapy [120].

### 4.2. The Controversial Strategies of Combined Therapy

However, numerous clinical studies have demonstrated that combining ICIs with some targeted drugs only brings about more side effects with no additional efficacy, such as combining ICIs with EGFR-TKIs, which suggests that tumors with driver gene mutation, such as *EGFR*, *ALK* and *ROS1* mutations and fusions, are not recommended for the combination of ICIs plus targeted drugs [4,72]. It may be because using EGFR-TKIs alone is able to make lung cancer with driver gene mutations get enough remission, and patients with *EGFR*, *ALK*, and *RET* mutations have a low response rate to ICIs because of the low PD-L1 expression [121,122,123]. Recent studies revealed that *EGFR/ALK*-positive NSCLC was significantly associated with the immunosuppressive environment and deficient CD8+ tissue-resident memory cells, contributing to the low efficacy of ICIs plus TKIs, and how to reverse the immunosuppressive phenotype would be a critical challenge [124,125]. The combination of ICIs and angiogenesis inhibitors plus chemotherapy has the potential to reverse the limitation of ICIs in NSCLC patients with *EGFR* mutations, which may be due to the stimulation of CD4/8+ lymphocytes and the depletion of immune suppressive cells with chemotherapy [122,126]. Therefore, we propose that ICIs combined with angiogenesis inhibitors plus low-dose chemotherapy may be a potential strategy for NSCLC patients with *EGFR* mutations. Despite the triple therapy of combining atezolizumab plus vemurafenib and cobimetinib being approved for advanced melanoma, the other two trials (KEYNOTE-022 and COMBI-i) of different agents in the triple-combination only observed modest improvement in PFS with high rates of side effects [94,95,96]. It is suggested that the effect of combination therapy is heterogeneous, dependent on the specific agents, and related to the side effects, even if these inhibitors target the same molecules.

### 4.3. The Side Effects of Combined Therapy

The spectrum of side effects from the combination of ICIs and targeted agents are a collection of side effects from ICIs and targeted agents monotherapy. The most common adverse events are tolerant and treatable, although different combined treatments lead to diverse side effects. It is observed that the occurrence rate of any grade adverse events is similar between monotherapy and combined therapy, while the occurrence rate of serious adverse events (≥grade 3) is higher in the combinations. For example, combining ICIs and angiogenesis inhibitors incurred more serious adverse events than using angiogenesis inhibitors alone, such as high-grade hypertension, decreased platelet count, and proteinuria, and there were more bleeding events in the combined group leading to the withdrawal of treatment than angiogenesis inhibitor monotherapy [47,50]. Although the incidence of treatment-related death was low in the combination of ICIs and angiogenesis inhibitors, it was reported that the percentage of the death due to hepatic failure and interstitial lung disease increased in the combinational context [50,61]. Immune-related adverse events including hepatic dysfunction, skin-related adverse reactions, and hypothyroidism were also increasingly induced through combining ICIs with targeted drugs than ICIs monotherapy [47,50,61,95]. Because of hepatic toxicities after using ICIs plus ALK inhibitors, the enrollment was closed and the related clinical trials were terminated early, which gives us a warning [110]. It reminds us to strengthen surveillance and intervention for those events when combination treatments are implemented [83,85].

### 4.4. The Predictive Biomarkers of Combined Therapy

In addition, the clinical data emphasize that the predictive biomarkers are still applicable when ICIs were combined with targeted therapy. It is well known that PARP inhibitors and BRAF-MEK inhibitors are approved in patients with germline *BRAC1/2* or *BRAF*^V600^ mutations, respectively, and these biomarkers are also crucial for combined therapy [95,127]. Immunotherapy biomarkers play important roles in combined therapy. The positive PD-L1 status was defined as greater than or equal to 1% of tumor-infiltrating immune cells staining PD-L1 expression in the tumor area, and PD-L1 inhibitor atezolizumab was approved with paclitaxel for injection by FDA in metastatic TNBC with PD-L1 positive. Combining ICIs and angiogenesis inhibitors improved the prognosis in RCC and NSCLC patients with positive PD-L1 expression, and more clinical benefits of combining ICIs with T-DM1 in patients with HER2-positive advanced breast cancer were only observed in the PD-L1 positive subgroup [54,60,76]. It was also shown that the combination of ICIs and PARP inhibitors for advanced NSCLC patients only improved the prognosis in PD-L1 positive group [88]. Although numerous examples of evidence have demonstrated that positive PD-L1 expression is an indispensable predictive biomarker for ICI therapy, PD-L1 could not predict the efficacy of ICIs in some studies, such as nivolumab in HCC (CheckMate 040) [128,129,130]. TMB and MSI are also potential predictors for ICIs, while studies included in this review cannot analyze the association because of few patients with available TMB/MSI statuses. It is suggested that higher TMB was associated with ICI response in NSCLC and melanoma, whereas it could not predict the efficacy of PD-1 inhibitors in RCC [131]. MSI-H status was related to the better response of ICI therapy, and pembrolizumab was approved by FDA for the treatment of metastatic MSI-H solid tumors. MSI-H status is rare in patients, and it also needs further study to clarify its value in combined therapy [132]. Some gene mutations and gut microbiota are novel biomarkers for immunotherapy. A study demonstrated that *TP53* mutations were associated with immune activation in NSCLC, and DNA damage repair gene mutations play an important role in upregulating the expression of PD-L1, which might be therapeutic targets for combination therapies with ICIs [133,134]. In addition, gut microbiota can regulate the response to immunotherapy, which leads to a new direction of immunotherapy. It was shown that higher taxa richness of fecal from HCC patients had better responses for ICI [135]. Based on the indications approved by the FDA, PD-L1, and MSI-H status were predictive biomarkers identified for ICIs, and it is important to explore and utilize proper biomarkers to help identify the potential patients who will benefit from the combination of ICIs with targeted therapy.

In conclusion, combining ICIs and targeted drugs showed a significant synergistic effect in some tumors. The combinational strategies are distinct for different tumors, so initiating combinational treatment of ICIs and targeted drugs should be carefully carried out. Looking forward to the future, ICIs combined with ADCs may be an innovative strategy for cancer treatment based on the results of the KATE2 study [76]. A continuous phase III trial, KATE3, investigating the efficacy of atezolizumab plus T-DM1 in PD-L1-positive, HER2-positive advanced breast cancer, is ongoing. Another phase II trial (NCT04468061) is recruiting TNBC patients for treatment by combining pembrolizumab and sacituzumab govitecan (a Trop2-ADC). ADCs will exert more important roles in cancer immunotherapy, which has the advantages of both antibodies and cytotoxic drugs. In addition, other immunotherapies may play an important role in the future, including CAR-T immunotherapy, NK cell, and Treg therapy.

## Figures and Tables

**Figure 1 cancers-15-02858-f001:**
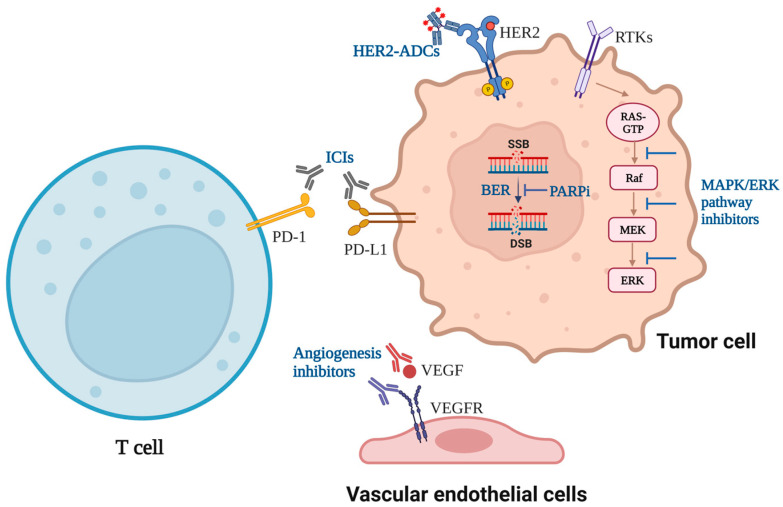
The current effective strategies of combining ICIs and targeted drugs. It is effective to combine PD-1/PD-L1 inhibitors plus angiogenesis inhibitors for advanced HCC, RCC, and EC, and the combination of ICIs and angiogenesis inhibitors plus chemotherapy has the potential to reverse the limitation of ICIs in NSCLC patients with *EGFR* mutations. PD-L1 inhibitor plus BRAF-MEK inhibitors for advanced melanoma with *BRAF*^V600^ mutations has been approved by FDA. PD-L1 inhibitor plus T-DM1 for PD-L1-positive, HER2-positive advanced BC, and PD-1/PD-L1 inhibitors plus PARP inhibitors in BC with germline *BRCA1/2* mutations and OC have potential clinical benefits. ICIs: immune checkpoint inhibitors; HER2-ADC: human epidermal growth factor receptor 2-antibody-drug conjugate; PARPi: Poly ADP-ribose polymerase inhibitor; RTKs: receptor tyrosine kinases; SSB: single strand break; BER: base excision repair; DSB: double-strand break; HCC: hepatocellular carcinoma; RCC: renal cell carcinoma; EC: endometrial cancer; BC: breast cancer; OC: ovarian cancer.

**Table 1 cancers-15-02858-t001:** Clinical Trials with Results of Combining ICIs and Targeted Drugs.

Disease	Phase	Name/No. (Setting)	Patient and Treatment	Results
Angiogenesis inhibitors				
Advanced HCC	III	IMbrave150 (1st-line)	501 (336 with Atezolizumab+Bevacizumab, 165 with Sorafenib)	mPFS: 6.8 vs. 4.3 ms; 1-year OS: 67.2% vs. 54.6%; ORR: 27% vs. 12%.
HBV-associated HCC	III	ORIENT-32 (1st-line)	571 (380 with Sintilimab+IBI305, 191 with Sorafenib)	PFS: 4.6 vs. 2.8 ms; OS: NR vs. 10.4 ms.
Advanced RCC	III	IMmotion151 (1st-line)	915 (454 with Atezolizumab+Bevacizumab, 461 in Sunitinib; 362 with PD-L1+)	mOS (PD-L1+): 38.7 vs. 31.6 ms; mPFS (PD-L1+): 11.2 vs. 7.7 ms.
Advanced RCC	III	JAVELIN Renal 101 (1st-line)	886 (442 in Avelumab+Axitinib with 270 PD-L1+; 444 in Sunitinib with 290 PD-L1+)	PFS (PD-L1+): 13.8 vs. 7.0 ms; PFS (whole): 13.3 vs. 8.0 ms.
Advanced RCC	III	KEYNOTE-426 (1st-line)	861 (432 with Pembrolizumab+Axitinib, 429 with Sunitinib)	OS: NR vs. 35.7 ms; PFS: 15.4 vs. 11.1 ms.
Metastatic NSCLC	III	IMpower150 (1st-line)	1202 (402 in ACP with 213 PD-L1+; 400 in ABCP with 209 PD-L1+; 400 in BCP with 195 PD-L1+)	mOS: 19.5 ms (ABCP) vs. 14.7 ms (BCP) and 19.0 ms (ACP); mPFS: 8.4 (ABCP) vs. 6.8 (BCP) and 6.3 ms (ACP);
Advanced endometrial cancer	II	NCT02501096 (≥2nd-line)	108 (11 with MSI-H, 94 with MSS) Pembrolizumab+Lenvatinib	mPFS: 7.4 ms; mOS: 16.7 ms; 2-year ORR: 63.6% (MSI-H) vs. 36.2% (MSS)
Refractory mCRC	II	BACCI (Multi-line)	133 (82 with Atezolizumab+Bevacizumab+Capecitabine, 36 with Bevacizumab+Capecitabine)	mPFS: 4.4 vs. 3.3 ms; 1-year OS: 52% vs. 43%; ORR: 8.54% vs. 4.35%
mCRC	II	AtezoTRIBE (1st-line)	218 (145 with Atezolizumab+Bevacizumab+FOLFOXIRI, 73 with Bevacizumab+FOLFOXIRI)	mPFS: 13.1 vs. 11.5 ms;
EGFR/HER2 inhibitors				
Advanced *EGFR*^T790M^ mutant NSCLC	III	CAURAL (2nd/3rd-line)	29 (14 with Durvalumab+Osimertinib, 15 with Osimertinib)	ORR: 64% vs. 80%; DCR: 93% vs. 100%.
Advanced HER2+ gastric cancer or GEJC	III	KEYNOTE 811 (1st-line)	264 (133 with Pembrolizumab+Trastuzumab+ChT, 131 with Trastuzumab+ ChT)	ORR: 74.4% vs. 51.9%;
Advanced SCC of the lung	II	LUX-Lung IO (2nd-line)	24; Pembrolizumab+Afatinib	mOS: 29.3 weeks; ORR: 12.5%
HER2+ advanced BC	II	KATE2 (2nd-line)	202 (133 in T-DM1+Atezolizumab, 69 in T-DM1)	PFS (whole): 8.2 vs. 6.8 ms; PFS (PD-L1+): 8.5 vs. 4.1 ms;
Advanced HER2+ gastric cancer or GEJC	I/II	CP-MGAH22-05 (Multi-line)	95; Pembrolizumab+Margetuximab	OS: 12.48 ms; PFS: 2.73 ms; ORR: 18.48%
Trastuzumab-resistant, advanced HER2+ BC	I/II	PANACEA (Multi-line)	52 (40 with PD-L1+, 12 with PD-L1-); Pembrolizumab+Trastuzumab	mPFS: 2.7 vs. 2.5 ms; ORR: 15% vs. 0%.
Poly (ADP-ribose) polymerase inhibitors				
HER2-negative stage II/III BC	II	I-SPY2 (1st-line)	372 (73 with Durvalumab+Olaparib+Paclitaxel, 299 with Paclitaxel)	pCR: 37% vs. 20%
Advanced TNBC or Recurrent OC	I/II	TOPACIO (Multi-line)	TNBC: 55; OC: 60; Pembrolizumab+Niraparib	TNBC: mPFS (g*BRCA1/2*m): 8.3 ms; mPFS (g*BRCA1/2*-wt): 2.1 ms OC: mPFS: 3.4 ms; ORR: 18%
Platinum-sensitive pancreatic cancer	I/II	Parpvax (≥2nd-line)	84 (44 with Niraparib+Nivolumab, 40 with Niraparib+Ipilimumab)	6-month PFS (Niraparib+Nivolumab): 20.6%; 6-month PFS (Niraparib+Ipilimumab): 59.6%
MAPK/ERK signaling inhibitors				
*BRAF*^V600^ mutated advanced melanoma	III	IMspire150 (1st-line)	514 (256 in Atezolizumab+Vemurafenib+Cobimetinib, 258 in Vemurafenib+Cobimetinib)	PFS: 15.1 vs. 10.6 ms; 5-year OS: 36% vs. 43%; mDOR: 21.0 vs. 12.6 ms.
*BRAF*^V600^ mutated advanced melanoma	III	COMBI-i (1st-line)	532 (267 in Spartalizumab+Dabrafenib+Trametinib, 265 in Dabrafenib+Trametinib)	mPFS: 16.2 vs. 12.0 ms; ORR: 69% vs. 64%.
mCRC	III	IMblaze370 (3rd-line)	363 (183 in Atezolizumab+Cobimetinib, 90 in Atezolizumab, 90 in Regorafenib)	mOS: 8.87 vs. 7.10 vs. 8.51 ms; mPFS: 1.91 vs. 1.94 vs. 2.00 ms.
*BRAF*^V600^ mutated advanced melanoma	II	NCT01673854 (1st-line)	70; Ipilimumab+Vemurafenib	mOS: 18.5 ms; mPFS: 4.5 ms.
*BRAF*^V600^ mutated metastatic melanoma	I/II	KEYNOTE-022 (Multi-line)	120 (60 in Pembrolizumab+Dabrafenib+Trametinib, 60 in Dabrafenib+ Trametinib)	mPFS: (16.0 vs. 10.3 ms); mDOR: 18.7 vs. 12.5 ms; mOS: NR vs. 23.4 ms.

HCC: hepatocellular carcinoma; mRCC: metastatic renal cell carcinoma; NSCLC: non-small cell lung cancer; mCRC: metastatic colorectal carcinoma; GEJC: gastroesophageal junction cancer; SCC: squamous cell carcinoma; BC: breast cancer; TNBC: triple-negative breast cancer; OC: ovarian cancer; HER2+: HER2 positive; T-DM1: trastuzumab emtansine; PD-L1+: PD-L1 positive; PD-L1−: PD-L1 negative; MSI-H: microsatellite instability-high; MSS: microsatellite stable; g*BRCA1/2*m: germline *BRCA1/2* mutant; g*BRCA1/2*-wt: germline *BRCA1/2* wild type; ABCP: Atezolizumab+Bevacizumab+Carboplatin+Paclitaxel; FOLFOXIRI: fluorouracil+leucovorin+oxaliplatin+irinotecan; ChT: chemotherapy: 5-fluorouracil+Cisplatin or Capecitabine+Oxaliplatin; PFS: progression-free survival; OS: overall survival; mPFS: median progression-free survival; mOS: median overall survival; ORR: objective response rate; ms: months; DCR: disease control rate; DOR: duration of response; mDOR: median duration of response; pCR: pathologic complete response; NR: not reached.

## Data Availability

All data generated or analyzed during this study are included in this published article.

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
