# Peer review of "Immune Checkpoint Inhibitors Combined with Targeted Therapy: The Recent Advances and Future Potentials"

_cancers, 2023, doi:10.3390/cancers15102858_

Round 1

Reviewer 1 Report

The review assesses a current, timely topic in medical oncology.
We recommend some changes:
- We believe this article is suitable for publication in the journal although major revisions are needed. The main strengths of this paper are that it addresses an interesting and very timely question and provides a clear answer, with some limitations. 

- The authors should further discuss predictors of response to immunotherapy. It is fundamental topic which should be expanded and better discussed. 
Immune checkpoint inhibitors (ICIs) including pembrolizumab, nivolumab, durvalumab, atezolizumab, etc. have been recently evaluated in cancer patients. Despite ICI seem to have finally found their role in several tumors as monotherapies or as part of combinatorial strategies, several questions remain unanswered. Among these, the lack of validated biomarkers of response represents an important issue since only a proportion of patients benefit from immunotherapy. Based on these premises, a greater understanding of the role of potential biomarkers including programmed death ligand 1 (PD-L1) expression, tumor mutational burden (TMB), microsatellite instability (MSI) status, gut microbiota and several others is fundamental. In addition, clinical trials on immunotherapy widely differed in terms of drugs, patients, designs, terms of study phases, and inconsistent clinical outcomes.The background of the changing scenario of cancer immunotherapy should be better discussed, and some recent papers regarding this topic should be included ( PMID: 35403533; PMID: 36368251; PMID: 34976841  ).

- An additional, schematic figure should be added. It would help readability.

Major changes are necessary.

Reviewer 2 Report

This review with therapeutic strategies for the combination of immune checkpoint inhibitors (ICIs) and targeted drugs is very meaningful and could be a very impactful article. However, the manuscript lacks many details to be a comprehensive overview especially on a topic as broad as this manuscription. Listed below are some of the suggestions.

1. The discussion of combination therapy mechanism and conclusion could not be comprehensive enough. It is suggested the immunological molecular mechanism of combination therapies and could be discussed in detailed. The conclusion for this review could be more selective and clarified, and the discussion part need to be divided into subunit.

2. It is suggested more research evidence should be provided that active angiogenesis may lead to immunosuppression and osmotic barrier (line 115-118).

3. The authors consider that chemotherapy is still indispensable in combinational treatment of ICIs and angiogenesis inhibitors in cancers (line 197-198). It is suggested the indications and precautions of chemotherapy should be discussed in detailed when combining ICIs and angiogenesis inhibitors.

4. The data in Table 1 is presented in confusion, please re-summarize it.

5. It is suggested more research evidence should be discussed in detailed that the activation of EGFR pathway is associated with immunosuppression (line 222).

6. Clinical studies of ICIs combined with EGFR or HER2 inhibitors only list the results of data in a certain extent. It is suggested systematic summary and conclusion should be discussed in detailed (line225-276).

7. It is suggested more research evidence should be discussed in detailed that PARP inhibitors involved in anti-tumor immunity to support combining PARP inhibitors will be a potential strategy (line 286-290).

8. It is suggested systematic summary and conclusion should be discussed in detailed by combining ICIs and PARP inhibitors (line291-319).

9. It is suggested more research evidence should be discussed in detailed that combining the ICIs with MAPK/ERK signaling inhibitors may induce a long time of disease response (line 334-346).

10. It is suggested systematic summary and conclusion should be discussed in detailed by combining ICIs and MAPK/ERK signaling pathway inhibitors (line 337-365).

11. It is suggested the mechanism and application prospect should be discussed in detailed for immunotherapy in hormone-dependent tumors (line 366-381).

Reviewer 3 Report

Majors:

I found Figure 1 to be poorly informative and oversimplified. For example, it should provide more details on targeted drugs, solid tumors, and immune checkpoint inhibitors (ICIs). Additionally, topographic information should be represented, and the human and mice pathways should be metioned. In my opinion, Figure 1 must be improved and Fig 1 is not necessary if it is similiar to table1. For instant, Table1 showing the effects of various ICIs on receptors is needed.

Furthermore, the review lacks data from 'immune' studies with these therapeutics. eg, T cells, NK, Treg?

As a perspective article, the authors should provide more of their personal interpretation of the literature and their views on where the field will move in the near future. This is lacking and reduces the impact of the manuscript in its current form. 

Round 2

Reviewer 2 Report

The authors could improve the depth of discussion part reperfectly,especially for necessary considerations and discussions on the future clinical research and diagnosis and treatment prospects.

Reviewer 3 Report

In accordance with Author replies, Figure 1 and Table 1 have been incorporated. We have also addressed point 3.

Point 2. Discussion of CART T cells, T cells and signaling, but not incorporated into the text? There would be interest in seeing this part in the author's review. This would include the combination of ICIs and targeted therapies, such as the CART T cell signaling section, and other immunotherapies. There is a need for further discussion of this part

Round 3

Reviewer 3 Report

my points have been addressed. Fig 1, it is still blur. Was your graphic tool being cited according under the license, did you subscribed ? it may be required to do for licensing purpose.